# Toward the Genetic Improvement of Drought Tolerance in Conifers: An Integrated Approach

**Paolo Baldi** [1] **and Nicola La Porta** [1,2,*]

1   Research and Innovation Centre, Fondazione Edmund Mach (FEM), 38098 San Michele all'Adige, Trento, Italy
2   EFI Project Centre on Mountain Forests (MOUNTFOR), 38098 San Michele all'Adige, Trento, Italy
*   Correspondence: nicola.laporta@fmach.it

**Abstract:** The constant rise in the global temperature and unpredictable shifts in precipitation patterns are two of the main effects of climate change. Therefore, there is an increasing amount of interest in the identification of tree species, provenances and genotypes capable of withstanding more arid conditions and tolerating drought stress. In the present review, we focus our attention on generally more susceptible conifers and describe the different strategies that plants adopt to respond to drought stress. We describe the main approaches taken in studies of conifer adaptations to low water availability, the advantages and limitations of each, and the main results obtained with each of these approaches in the recent years. Then we discuss how the increasing amount of morphological, physiological and genetic data may find practical applications in forest management, and in particular in next-generation breeding programs. Finally, we provide some recommendations for future research. In particular, we suggest extending future studies to a broader selection of species and genera, increasing the number of studies on adult plants, in particular those on gene expression, and distinguishing between the different types of drought stress that a tree can withstand during its life cycle. The integration of data coming from different disciplines and approaches will be a key factor to increasing our knowledge about a trait as complex as drought resistance.

**Keywords:** gymnosperms; molecular; next-generation breeding; water stress; genetics; genomic; drought resistance; *Picea*; *Abies*; *Pinus*

## 1. Introduction

According to the FAO's Global Forest Resources Assessment 2020 (https://www.fao.org/documents/card/en/c/ca8753en (accessed on 2 July 2022)), 4.06 billion hectares of land are covered by forests worldwide, corresponding to over 30% of the total land area. Over the last decades, climate change has increased the risk of drought stress (DS) in many regions, mainly via increased temperatures, altered precipitation patterns and faster snow melt [1,2]. The effects of droughts on forest plants vary, from reduced growth in cases of moderate droughts to mass mortality if severe droughts occur [3,4]. Global simulations predict widespread massive tree mortality under the projected rise in global temperatures and extremes that accompanies drought [5]. Therefore, it is of primary importance to understand plant response and adaptation mechanisms to drought in order to properly manage tree populations and select those individuals or provenances that show a higher resistance level. Drought resistance is a very complex trait with both environmental and genetic components, and different populations of the same species may not respond equally to a given climate [6,7]. Moreover, several distinct aspects can be important when drought tolerance (DT) is concerned. As an example, in regards to wood production, the capability of the plant to grow in low-water conditions might be the most important trait to consider, while for a natural forest, adaptations to highly variable water availability might be more favorable. In all cases, the main target of research studies is to understand what traits are more important to achieving drought resistance and to find the relationship between phenotypes and genotypes [8,9].

Conifers originated more than 300 million years ago and currently dominate many temperate and boreal forests. They have very large genomes (18 to 35 Gb), showing a very different structure and composition when compared with those of angiosperm genomes. Despite an apparently conserved genome structure, conifers demonstrate some competitive capacity, as different taxa are adapted to a wide variety of environmental conditions [10]. Even though angiosperms and gymnosperms share some general principles regarding drought resistance, there are also significant differences. Some studies suggest that angiosperm tree species, in general, tend to be less sensitive to drought than gymnosperm species [11,12]. At first glance, one of the fundamental differences in water transport is the size and function of conifer tracheids and angiosperm vessels [13]. In fact, even though conifers have greater stem hydraulic safety than angiosperms, during drought events, conifers experience more frequent embolisms than angiosperms in distal tissues [14]. Embolism repair is likely driven by sugars that come from nearby parenchyma cells, and conifers have few carbohydrates or parenchyma in their xylems compared to angiosperms. So, even though conifers tend to experience embolism more frequently in their leaves and roots than angiosperms, these organs may act as hydraulic circuit breakers to prevent stem embolism in conifers [15]. Recent ecophysiological studies have suggested the existence of contrasting strategies in angiosperms and gymnosperms regarding traits related to DT, such as the accumulation of non-structural carbohydrates (NSCs) [16–18]. Other studies reported that a reduction in precipitation is predicted to have negative effects on the growth of both angiosperms and gymnosperms, while increased temperatures may result in a performance disadvantage for conifers [19]. Gymnosperms, due to their long life cycle (most trees older than 1000 years are gymnosperms), have a high chance of experiencing numerous drought events during their lives [20]. Moreover, even drought-resistant evergreen gymnosperms growing in Mediterranean forests might be strongly impacted if drought periods become more frequent and severe [21]. The present review is focused on conifers and will not describe in detail all the numerous differences between gymnosperms and angiosperms. Such differences have been extensively reviewed elsewhere [22–25].

In the next paragraphs, we summarize the main approaches that have been used over the last few years to study a complex trait, drought resistance, in conifers. First of all, we define the main aspects of the trait and the different strategies that trees can adopt to increase their DT. Then we discuss some of the more frequently used techniques and review the main findings that have been obtained so far. Finally, we describe how the results can be applied in forest management, tree improvement programs and future research.

## 2. Plant Strategies to Cope with Drought

Intuitively, a very simple definition of DT is the ability of a plant to survive a prolonged period of water shortage. This can be true in particular environments, such as deserts or semi-arid regions where water availability is low most of the time. For the analysis of DT in more temperate areas, different climatic drivers and levels of variability among tree species should be taken into account [26]. Among other factors, trees' growth and in particular trees' short-term responses to extreme drought events are two of the most frequently studied [27]. Plants that are adapted to low water levels can also show a very low growth rate in favorable conditions as a consequence of more conservative resource usage, and therefore they may not be competitive with other, less conservative species [28,29]. Moreover, some studies on *Abies concolor*, *Pinus lambertiana* and *P. sylvestris* L. have correlated slow growth and sudden decreases in growth with a higher mortality [30,31].

Therefore, growth plasticity, the capability of plants to display a high growth rate in good climatic conditions and a low growth rate in drought conditions, can be considered a very favorable trait, especially in variable environments [32].

Different types of DS actually exist. A single extreme episode of drought is a different type of stress compared to the stress resulting from low but constant water availability. Moreover, a different kind of stress results from multiple drought episodes occurring during a growing season, and even a single drought event can occur in different periods of

a plant's growing season. Therefore, plant response is very complex, and multiple strategies should be adopted to cope with drought. Such strategies can be divided into three main groups: drought avoidance, drought resistance and drought resilience (Figure 1).

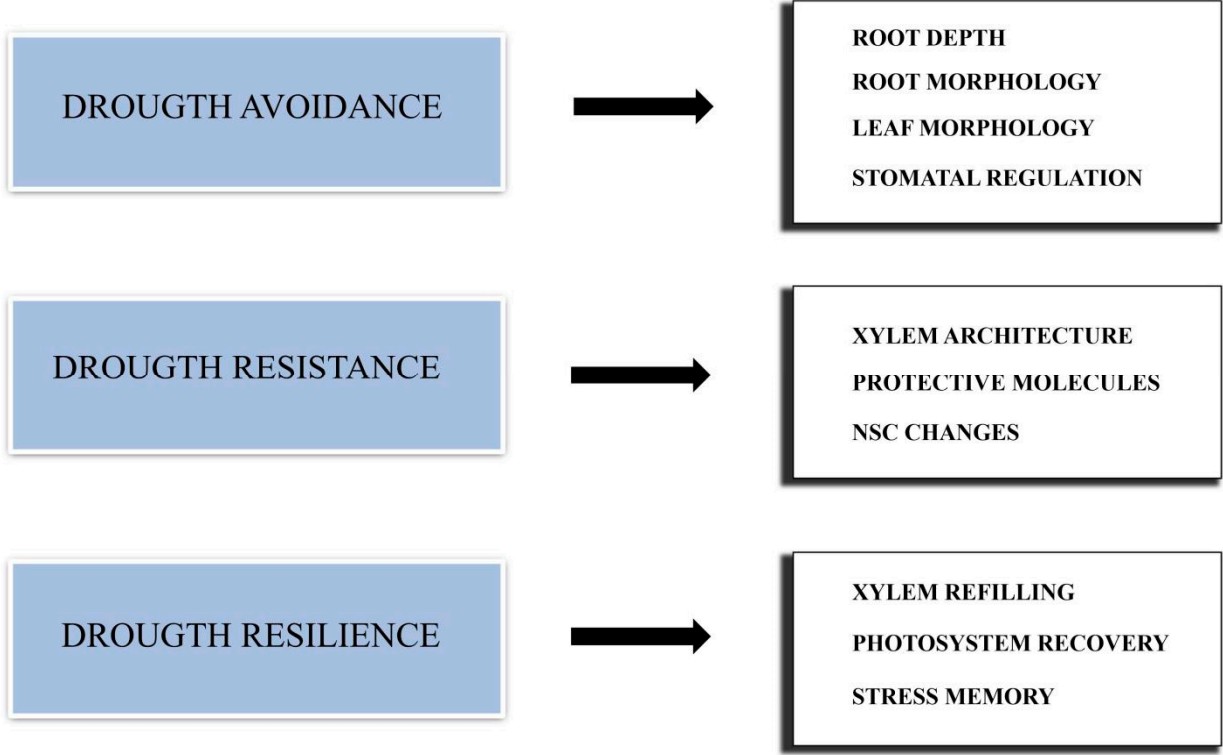

**Figure 1.** Schematic representation of the main strategies adopted by conifers to cope with drought stress.

### 2.1. Drought Avoidance

Drought avoidance is the capability of a tree to minimize water loss and maximize water uptake [33,34]. Avoidance mechanisms are generally characteristic of a species and act at the whole-plant- or organ-level. Plants with a deeper root system are more efficient in reaching subsoil water [35] and less prone to experiencing water stress during the growing season than plants with shallow roots. Root depth can also influence a plant's hydraulic status. For several conifers (*Abies lasiocarpa*, *Picea engelmannii*, *Pseudotsuga menziesii*), different studies have indicated that deeper water sources, mostly derived from snow melt, are primary reservoirs to support forest water demand [36,37]. Roots with a small diameter and a high ratio between their surface area and volume may present a lower susceptibility to cavitation and therefore an increased drought resistance [38]. On the other hand, drought can induce a severe dieback of the fine roots in *Larix decidua*, leaving the plant with a very small amount of live roots [39], which in turn can limit the plant's response to precipitation pulses [40]. In Norway spruce, even a moderate drought was shown to modify the total biomass and the depth distribution of its fine roots [41]. In this species, an extended drought induces the deeper growth of fine roots and fungal hyphae [42].

Root and branch patterns can also influence the timing and severity of water stress, while the total numbers of branches and leaves have an effect on plant transpiration. Conifers can respond to water stress by modifying their needle morphology. In *Picea abies*, the needle morphology was shown to be affected by DS, but only in the portions of the canopy exposed to sunlight [43]. Another study found a reduction in the needles' tracheid dimensions, cross-sectional area and xylem area [44]. A strategy to reduce the leaf area index and consequently transpiration is to induce branch die-back during severe

droughts [45]. As the water availability decreases, conifers can induce stomatal closure by two distinct mechanisms: isohydric trees use abscisic acid (ABA) as a messenger to induce stomatal closure [46], while anisohydric species, such as many Cupressaceae and some Taxaceae, let the water potential drop and use it as a signal for stomatal closure [33].

## 2.2. Drought Resistance

When drought avoidance strategies are not enough to overcome a period of low water availability, plants need to cope with DS [34]. Drought resistance is the ability of a plant to withstand such stress, with a series of characteristics that can be constitutive or induced. As an example, the xylem architecture can deeply influence the drought resistance of trees. Tracheids with a larger lumen are more prone to hydraulic failures [47,48]. In fact, conifer species with a higher hydraulic conductivity and photosynthetic capacity were found to be more sensitive to DS and tended to show weaker growth resistance to extreme drought events [49,50]. In contrast, more lignified tracheid walls and smaller inter-tracheid pits can increase their DT [51,52]. The lumen width and cell wall thickness of tracheids were found to vary throughout different seasons in Norway spruce, with those produced in dry seasons being thicker and having a smaller lumen [53,54]. These variations can affect trees' life and growth for several years, as the xylem, once produced, usually remains functional for multiple seasons [55]. Water loss can cause multiple damages at the cellular level, such as decreased cell turgor, protein denaturation and changes in membrane fluidity [56]. In order to reduce cellular damage, plants can synthesize protective molecules, including some proteins, such as chaperonins and dehydrins [57], the amino acid proline [58], various carbohydrates [59] and isoprenoids [60]. These compounds act as osmotically active molecules, contributing to the increase in cell turgor when the water potential is low. They can also stabilize cellular proteins and prevent membranes from leaking [61,62].

NSCs mainly formed by starch and soluble sugars, in addition to protecting cells from dehydration, are an important source of energy for metabolic processes [63]. Changes in NSCs have been reported for conifers during DS, according to modalities that can be different from species to species [64–66]. The exact meaning of such changes and their relationship with plant growth and mortality during DS are still unclear [67–70].

Another consequence of prolonged water deficits is oxidative stress, due to the accumulation of reactive oxygen species (ROS) that in turn can damage important cellular components, such as the photosystem [71]. To cope with that, conifers can produce antioxidant molecules, such as terpenes. During DS, terpene emissions generally increase, even though the response can be modified by severe water stress and the co-occurrence of other stresses [72].

## 2.3. Drought Resilience

Once the water availability is restored, plants can resume normal growth. The efficiency and speed of this process can be defined as drought resilience, and it is particularly important when wood production is considered [73]. There are several mechanisms that can influence drought resilience. Refilling the xylem following cavitations is one of these mechanisms. Active refilling has been observed for some conifers [74,75], even through water absorption via the needle's cuticle [76,77]. Nevertheless, multiple cycles of damage and repair may result in conduits that are more vulnerable to embolisms than those freshly produced by the vascular cambium [78,79]. Such a condition is known as cavitation fatigue [49,80]. The fast recovery of the photosystem's efficiency is another characteristic that can deeply influence a plant's resilience. This is particularly true for conifers, which are usually evergreen and therefore can be exposed more often to unfavorable water conditions than plants that go dormant [71]. Finally, a sort of stress memory seems to be present in plants. For instance, plants that are exposed to repetitive water stress can maintain a smaller stomatal aperture than that of unstressed ones, even when re-watered [81,82]. Stress memory is thought to rely on post-transcriptional and epigenetic mechanisms and may represent an advantage for plants, especially in highly variable environments [83,84].

## 3. Studying Drought Tolerance in Conifers

In a context of climate change, in which drought events are predicted to increase in frequency and severity in the near future [85], it is necessary to understand how plants are capable of adapting to low water availability. This is particularly true for conifers, which are often a dominant component of arid zone forests. Plant adaptations to drought can be studied at different levels. The main approaches consider traits at anatomical/morphological, physiological/biochemical and genetic levels.

### 3.1. Anatomical and Morphological Studies

Morphological studies usually link a specific phenotype to DT. In conifers, tree growth, often measured by the basal area increment or tree ring width, is a widely used parameter that can be correlated to meteorological data to assess plants' responses to drought events [86]. Tree growth measurements have been applied successfully to understand the effects of forest management on the drought responses of several species, such as *Pinus nigra* [87], *Pinus halepensis* [88] and *Pinus sylvestris* [89,90]. The basal area increment was coupled with shoot elongation measures to compare the drought responses of adult trees and saplings in three different *Pinus* species, in order to predict future scenarios of relic forests under climate change [91]. In another study, the stem radial growth was used to compare the responses of *Pinus edulis* (isohydric species) and *Juniperus monosperma* (anisohydric species) to DS [92]. Tree height can be used as a morphological characteristic to study conifers' response to drought. In a natural forest, the relationships between dominant trees and smaller ones may vary with climate change. When an ample supply of water is guaranteed, light can be the main limiting factor for plant growth, so the dominant trees are favored. In contrast, when the water availability becomes a limiting factor, the more shaded positions of smaller trees may limit transpiration and therefore compensate for the lack of light [93].

Another important morphological characteristic that is associated with drought resistance is the xylem morphology, in particular the lumen diameter and the thickness of the tracheid walls. These characteristics largely influence the xylem's water transport efficiency as well as its safety, which influence the probability of tracheid implosions during water stress [94]. Overall, conifers are capable of modifying the xylem's structure in response to droughts [95]. An arid climate seems to promote xylem efficiency over safety [79,96]. In a recent study on *Picea abies*, a macroscopic characteristic, namely stem cracks, was associated with water stress [97]. In the same study, the thickness of the tracheid walls, rather than the lumen diameter, was found to be the main anatomical characteristic associated with tracheid collapses. This finding is particularly important for all tree species exploited for wood production, as the cell wall thickness was shown to have quite a high heritability [98]. Cell wall thickness is a parameter determining wood density, and wood density has been negatively correlated to growth in *Picea abies* [99]. The xylem phenology was also studied in conifers in relationship to drought [100]. Characteristics, such as cell differentiation, cell enlargement and cell wall thickening, seem to be influenced by water availability in several species, such as *Abies alba*, *Pinus sylvestris* [101] and *Larix decidua* [102].

When studying DT in plants, one should take into account that natural populations are not homogeneous but present a certain degree of genetic variation. Morphological traits, such as the basal area increment or tree ring width, can be used to assess the inter- and intraspecific genetic variation. This is usually performed in so-called provenance or common garden studies, in which seedlings from many different regions can be planted and studied in a common environment [103,104]. Provenance studies can be conducted across multiple sites or using multiple treatments in order to estimate the plasticity of traits [105]. The intraspecific growth response to drought was studied in *Abies alba* using provenances from Bulgaria, Italy, Romania and Czech Republic. The ring width, earlywood width and latewood width were measured and correlated to drought events over a period of over 20 years, in order to find provenances combining a high productivity and drought resistance [106]. In a second study, 43 populations of *Picea glauca* were evaluated for

DT by measuring a series of tree-ring traits. A significant genetic variation was found among populations in response to DS. In particular, the authors found that populations from drier geographical origins showed a higher resilience to extreme drought events when compared to populations from more humid geographical origins, indicating local genetic adaptations [107]. Similar experiments were performed using several phenotypical traits on different conifer species, such as *Pinus pinea* [108], *Pinus ponderosa* [109], *Pinus sylvestris* [110] and *Picea abies* [111]. An interspecific study was performed comparing the drought responses of *Picea abies*, *Abies alba*, *Larix decidua* and *Pseudotsuga menziesii*. This latter species had the highest drought resistance, while *Abies alba* had the best drought recovery. Nonetheless, even the most drought-sensitive species, *Picea abies* and *Larix decidua*, showed significant genetic variation within and among populations along their natural geographic areas, enough to justify targeted tree breeding and supportive forest management [112].

### 3.2. Physiological and Biochemical Studies

Most of the studies using morphological data to assess DT are focused on tree growth. This parameter is a good indicator of plant resilience, and it is very useful when wood production is the final goal. Nevertheless, DT is a very complex trait involving a number of different mechanisms. In order to identify such mechanisms and use them to highlight differences in plant responses to drought, physiological data (Table 1) can be used [113,114]. One of the most studied physiological processes related to water stress is photosynthesis [115–117]. In particular, the concentration of chlorophyll *a* can be obtained via the extrapolation of the emission of refracted light from foliage [118], therefore it is a parameter that can be measured by remote-sensing tools and can be used to monitor large areas [119,120]. Drought can damage Photosystem II (PSII), resulting in changes in fluorescence parameters [121]. The fluorescence measurement was shown to be a very sensitive proxy for DS [122], allowing researchers to assess physiological disturbances even before the appearance of visible symptoms [121]. In the last few years, several studies on conifers have been published [123,124]. In *Picea abies*, the chlorophyll *a* concentration and fluorescence parameters were measured together with the tree height to assess seedling performances under water stress. Both physiological parameters were good indicators of plants' drought sensitivity, even though differences were found depending on the soil type [125]. In *Abies alba*, the chlorophyll *a* fluorescence was measured, testing five provenances from different altitudes under mild water stress. Significant differences were found, with provenances from higher altitudes showing better performances under both optimal and low-water conditions, suggesting that there were local adaptations to drought and that fluorescence parameters can be applied during plant selection for resistant seedlings [126].

Another important physiological parameter that has been often used to assess plant DT is water use efficiency (WUE), which is the ratio between the carbon fixed by photosynthesis and the water loss [127–129]. Nevertheless, caution must be taken when using WUE as an indicator for DT, because it depends on different mechanisms, such as photosynthesis and transpiration, which can both vary and not always in the same way.

Moreover, several measures of WUE exist and even if they are often correlated, they are not interchangeable [130], so they may lead to contrasting results. One of the most frequently used methods for measuring WUE is the carbon ratio $\delta^{13}$ C [131]. In conifers, some studies have found higher $\delta^{13}$ C in populations originating from drier sites [28], while others have shown the opposite [132]. In *Pinus halepensis*, individuals from dry sites showed a lower WUE plasticity than those from mesic sites [128], while in another study, a higher average WUE was shown in individuals from drier sources [127]. Finally, it must be noted that, especially if measured during the whole growing season, high water usage due to highly plastic growth can reduce the WUE, even though plastic growth can be considered a desirable characteristic in moderately dry climates [28]. For all these reasons, it is always advisable to integrate WUE with other parameters when dealing with drought resistance [49,133]. As an example, in *Pinus mariana*, changes in morphological (shoot volume,

leaf mass area), physiological (respiration rate, light-saturated photosynthesis) and biochemical (osmolality, NSC) parameters were considered to quantify the impact of temporary drought and re-hydration on its internal carbon dynamics and shoot development [49,134]. More in general, biochemical data can be very useful to integrating physiological studies and understanding conifer responses to DS. Osmolytes, such as proline, chlorophyll *a* and *b*, and antioxidants, such as anthocyanins, carotenoids and phenolic compounds, can be measured and correlated to water availability [62]. In *Larix decidua*, six Carpathian populations were submitted to a month of mild DS under controlled conditions. Several biochemical parameters were measured, looking for the optimal biochemical markers for an early detection of drought symptoms [135]. A lot of attention has been given in the last decade to NSC dynamics, specifically to understanding whether a dehydration-induced carbohydrate limitation (source limitation) or a direct effect of low water availability (sink limitation) is responsible for reducing plant growth under DS [66,75,136]. Some indications seem to suggest that growth is not carbon-limited by drought as growth reductions may occur along with carbohydrate storage increases [68,137]. On the other hand, carbon reserves could be formed at the expense of growth as a strategy to prevent carbon starvation, a very dangerous condition for plants [138]. Moreover, in cases of longer and more severe droughts, a reduction in both the carbon storage and growth was observed [139].

**Table 1.** Physiological traits relevant for drought stress.

| Plant Traits | Species | Relevant Effects | Modulation under Stress | References |
|---|---|---|---|---|
| Photosynthetic activity | *Pinus* spp., *Picea* spp., *Pseudotsuga menziesii*, *Abies alba*, | Modulation of Calvin cycle and photosystem damage. | Reduction under stress. | [50,114,118,119, 122,123,125,126] |
| Carbon allocation and water-use efficiency | *Pinus* spp., *Picea mariana* | Compensation of reduced photosynthesis by increased remobilization. | Growth reduction for carbon investments in the maintenance of respiration and osmoprotection. | [18,28,49,68,127, 132,134,137] |
| Stomatal conductance | *Picea abies, Larix decidua, Pinus radiata* | Water consumption and storage. Leaf temperature regulation. | Increased stomatal resistance may help to prevent the xylem embolism. | [46,82,117] |
| Xylem cavitation and embolism | *Pinus* spp., *Picea* spp., *Abies* spp., *Juniperus* spp., *Pseudotsuga menziesii*, *Larix decidua*, *Taxus baccata* | Cavitation. Hydraulic conductivity. Growth capacity. Drought-induced mortality. | Isolation of embolized tracheids. Changes in carbohydrate contents. Osmotic adjustments. | [14,48,78,79] |
| Phenological phases | *Pinus sylvestris, Abies alba, Larix decidua* | Growth effects. Production and morphology of latewood tracheids. | Reduced growth. Phenological shifts to compensate for DS. | [100–102,133] |
| Root morphology and depth | *Picea abies, Larix sibirica* | Higher/lower tapping of soil water resources. | Reduced root biomass, increased root/shoot ratio, fine-root production decreased. Root depth increased. | [39,41,42] |
| Protection against active oxygen species | *Pinus* spp., *Picea* spp., *Pseudotsuga menziesii, Larix laricina, Abies balsamea* | Oxidative stress | Production of protective molecules. | [57–60,72] |

*3.3. Molecular Genetics Studies*

To further investigate drought responses and link morphological and physiological traits to the genes and/or genomic regions that are responsible for these characteristics, molecular genetics can be used [140]. One of the most applied approaches is a gene expression analysis. Although there are a lot of techniques to study gene expression, in the most recent publications, whole-transcriptome approaches [141–146] and/or quantitative real-time PCR (qRT-PCR) [143,144,147] have been used more frequently. In contrast to RNAseq and other whole-transcriptome techniques, qRT-PCR is very sensitive but in most cases it is used to study one or a few specific target genes [148] or to confirm the gene expression results obtained with less sensitive methods [144,149]. Via expression analyses, a range of genes have been identified that might be involved in the drought responses in conifers (Table 2).

Genes related to signaling and transcription factors, including AP2/ERF, bZIP, TCP, WRKY and MYB, have been found to be regulated during water stress in several conifer species, such as *Larix kaempferi* [150], *Pinus massoniana* [151], *Abies alba* [149] and *Pinus taeda* [152]. All these genes, due to their regulatory function, are usually expressed quite early during DS and therefore they could be considered good indicators of an efficient plant response. Nevertheless, this is not always the case. In *Pinus pinaster*, it was found that tolerant individuals can be pre-adapted to cope with drought, constitutively expressing stress-related genes; in contrast, in more sensitive individuals, these are induced by the onset of stress [142]. So, care must be taken when considering gene expression, and distinguishing between the different drivers of observed differences is of primary importance. Another important factor that must be considered when studying gene expression during drought is the type of treatment used. In some studies, DS was induced by stopping irrigation [145]; in others, a chemically-induced stress was used [153]. In some cases, the water was withheld for a given period [153]; in others, it was withheld until the needles wilted [152]. Thus, there is always the possibility of methodological artifacts [154]. Abscisic acid (ABA), a plant hormone involved in stomatal closure, shoot growth and water uptake, was shown to regulate many structural genes in conifers [144,145,151] even though ABA-independent pathways also exist in many species [155]. Other genes regulated during drought include those encoding for antioxidants [141,150], protective molecules, such as late embryogenesis abundant (LEA) proteins, which are thought to stabilize proteins and membranes [156], genes involved in lignin and sugar biosynthesis [150], flavonoid and terpenoid biosynthesis [141], aquaporins, which can affect the water permeability of membranes [151] and even pathogen resistance genes, such as nucleotide-binding, leucine-rich repeat proteins [157,158].

All the genes identified by expression analyses may have a role in drought responses, but it is not always easy to link a gene's expression, which is specific for the given time point in question, to more complex morphological and physiological traits, which are the result of different processes acting over a longer period of time. To do so, molecular markers can be used for quantitative trait locus (QTL), genome scan and genotype association studies. All these approaches aim at linking genes or genomic regions to a trait. Classic QTL studies exploit segregating populations to identify genomic regions linked to a polygenic trait but have had very limited success in conifers [130,159,160]. The main reason is the huge amount of space and time necessary to cross parental individuals and obtain enough progeny. Moreover, conifers have very large genomes, usually with a low linkage disequilibrium (LD), so high-resolution maps are needed for QTL detections and candidate gene identification [161]. The development of high-throughput systems, such as next-generation sequencing and single nucleotide polymorphism (SNP) arrays, is gradually overcoming such problems, allowing for association-mapping studies that exploit evolutionary recombinations in natural populations [162]. Nonetheless, until very recently, studying the molecular basis of heritable trait variation was still considered quite challenging in conifers and most of the studies were focused on a limited number of candidate genes [163–168]. This approach, although effective, is very limiting for the identification of new loci. To

overcome this problem, genome-wide association studies (GWASs) can be used. As an example, a GWAS was performed on *Sequoiadenron giganteum* and *Sequoia sempervirens*, trying to link 10 drought-related morphological and physiological traits to SNP markers. Seventy-eight new marker x trait associations were found in *Sequoia sempervirens* and six in *Sequoiadenron giganteum*, identifying a wide range of candidate genes involved in different biological processes, such as signaling, stress response and growth [169]. The genome size of conifers, which can range between 8 and 34 Gb [170], may still represent a problem. With some techniques, tens of thousands of SNPs can be obtained, covering most of the genome [171–173]. Nonetheless, most of such SNPs will be in noncoding regions. This, on one hand, is good for the identification of regulatory regions, but on the other hand, it may limit the number of gene associations detected. With the latest advances in sequencing and bioinformatics, it is possible to limit SNP analyses to the protein-coding region of genomes, the so-called exome [174]. With such an approach, the complexity of the conifer genome can be massively reduced, and even though the exome represents only a small fraction of the total genome, gene association is greatly facilitated [175]. Examples of exome-derived SNP association studies exist for several conifer species, such as *Pseudotsuga menziesii* [176] and *Pinus taeda* [177].

**Table 2.** Main categories of genes related to drought responses in conifers.

| Main Category | Subcategory | Species | Reference |
|---|---|---|---|
| Protective molecules | Dehydrins | 5, 9, 10, 11 | [141,148,149,152,156] |
| | LEA | 5, 9, 10, 11 | [141,149,152,156] |
| | HSP | 5, 11 | [152,156] |
| | Antioxidant | 1, 5, 9, 11 | [141,145,146,151,152,156] |
| | Other | 1, 5, 10, 11 | [146,149,151,152] |
| Gene transcription and signalling | TF | 1, 2, 3, 4, 5, 9, 10, 11 | [141,144–146,149–152,156,164] |
| | Hormones | 1, 2, 3, 5, 7, 9, 10, 11 | [141,144–147,149–152,156] |
| | Protein kinases | 2, 3, 5, 9, 10 | [141,144,149,150,156] |
| | Other | 2, 5 | [144,146,156] |
| Metabolic activity | Sugar | 1, 3, 4, 5, 9, 10, 11 | [141,145,149,150,152,156,164] |
| | Lipid | 1, 5, 9, 10 | [141,145,149,156] |
| | Cell wall | 5, 9, 10, 11 | [141,146,149,152] |
| | Flavonoids and phenylpropanoids | 1, 3, 4, 7, 9 | [141,145,147,150,164] |
| | Other | 1, 9 | [141,145] |
| Protein degradation | | 5, 9 | [141,156] |
| Photosynthesis | | 1, 5, 6, 7, 9 | [71,141,145,146] |
| Transport | | 1, 5, 9, 10, 11 | [141,146,149,151,152,156] |
| Biotic stress | | 1, 4, 5, 8, 10, 11 | [149,151,152,156,158] |

**Species**: 1: Pinus massoniana; 2: Pinus tabuliformis; 3: Larix kaempferi; 4: Picea glauca; 5: Pinus pinaster; 6: Picea abies; 7: Pinus sylvestris; 8: Pinus koraiensis; 9: Pinus halepensis; 10: Abies alba; 11: Pinus taeda.

## 4. Next Generation Breeding

The increasing amount of morphological, physiological and molecular data on conifer drought resistance can have practical applications in improving breeding programs [178–181] (Figure 2).

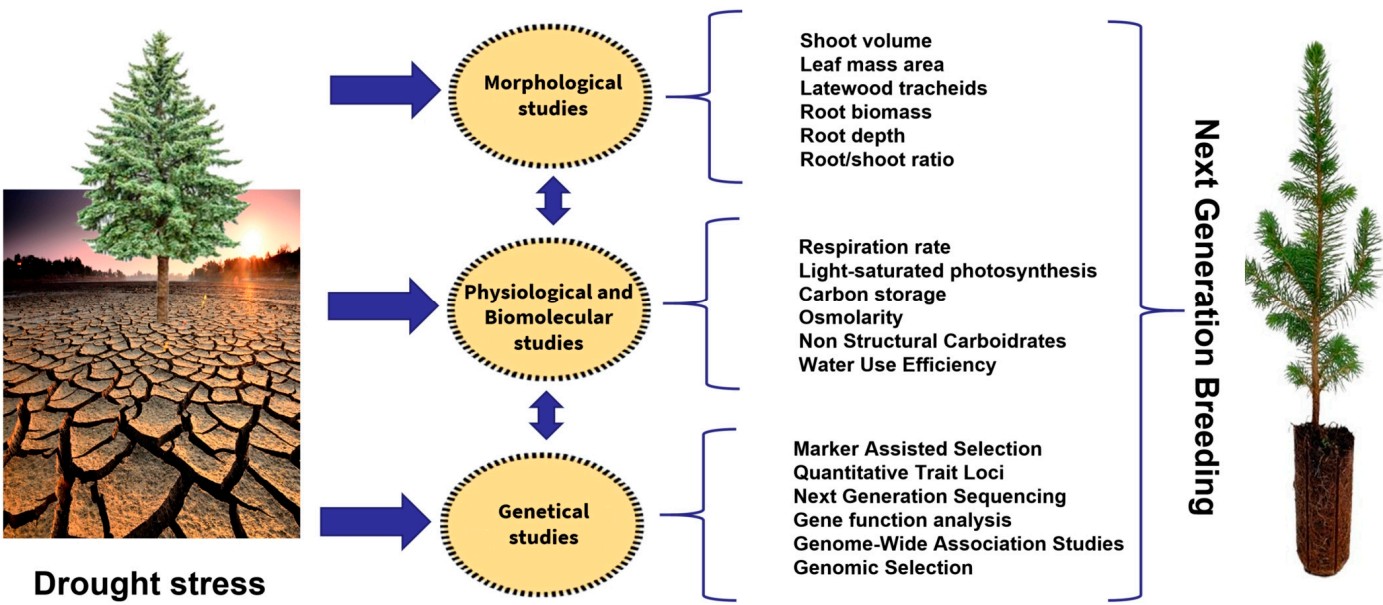

**Figure 2.** Next-generation breeding processes to improve drought stress in conifers.

As a matter of fact, traditional breeding programs for long-living trees, such as conifers, face a number of challenges. The most important is the great amount of space and time necessary to obtain significant achievements. Even though several breeding programs started in the 1950s, many of them are only in their first or second cycle of selection, with the most advanced ones, such as the loblolly pine (*Pinus taeda* L.) breeding program in the USA, in the third or fourth cycles [182]. Moreover, due to the high costs, tree breeding programs have been developed mainly for commercially important species and only recently has it become evident that it is important to also protect natural forests, which are often threatened by climate change [183]. In cases of drought, plant resistance or tolerance is controlled by multiple genes, each with a minor additive effect, so it is particularly challenging to achieve and maintain the desired level of resistance [184].

Another important factor that must be taken into account, especially when considering natural forests, is genetic diversity. Resistant populations developed by breeding programs have to maintain all the adaptive traits typical of the species or provenance, such as growth rate, cold resistance and pathogen resistance. Therefore, it is necessary to exclude the presence of negative correlations between drought resistance and other adaptive traits [182].

New emerging technologies can be efficiently used to link plant genome to phenotypic data and therefore could be used to improve the efficiency of traditional breeding and accelerate the selection of valuable genotypes. Nowadays, high-throughput technologies can be applied not only to genotyping but also to phenotyping, improving both data acquisition and analytical pipelines, as well as possibly leading to an unprecedented revolution in the way we have studied agriculture till now [185]. As an example, a number of portable and user-friendly chlorophyll fluorimeters have been introduced in the last few years. These, together with more sophisticated high-throughput hyperspectral imaging systems, have enhanced the accuracy of chlorophyll measurements. In this way, large phenotyping platforms have been implemented with an automatic control and data analysis system, allowing researchers to keep a large number of plants in parallel under constant monitoring, over long periods of time and in different environmental conditions [186]. These high-throughput phenotyping technologies are generally employed in controlled environments, such as growth chambers and greenhouses, so there are some limitations when predicting the performance of a plant under field conditions. Nonetheless, these systems can be very useful in studying how phenotypes change among different genotypes under uniform stress conditions [187].

A large amount of genomic resources is available for conifers. Despite their size and complexity, several conifer genomes have been sequenced [188–193]. A complete list of genomic resources of conifers, with the year of release, the Genebank storage and the accession number, has been reported by Traversari et al. [194]. These data, together with transcriptome studies, have provided useful information for the development of high-density genetic maps [195–199] and SNP arrays [200–204], which could be used in next-generation breeding approaches. To date, marker-assisted selection (MAS) has shown only a limited application in forest breeding [205,206]. The main reason is that MAS is quite reliable and easy to use in cases of monogenic characteristics, but it is much harder to be applied in cases of more complex traits, such as drought, as several markers linked to genetic loci involved in the control of the resistance must be identified. As already stated (see previous paragraph), classic QTL mapping had little success in conifers for the identification of genetic loci linked to traits of interest. Moreover, each locus can control a relatively small percentage of the resistance, so several loci should be screened at the same time in order to reach the desired level of resistance and should be validated using different populations before being routinely used for assisted selection [104,207]. Overall, it was shown that QTLs are not effectively suitable for MAS in forest trees, as they do not explain enough genetic variations for complex traits, such as drought resistance [208]. To date, one of the most promising approaches for next-generation breeding is genomic selection (GS) [209–211]. GS typically uses a large genome-wide panel of molecular markers (usually SNPs) whose effects on the desired trait are estimated in a so-called training population, which in forest trees is generally composed of one to a few thousand individuals. The molecular markers are used to build prediction models that can be later applied to any candidate resistant tree for which only the genotype is known, while the phenotype can be predicted by the models [206]. This is an obvious advantage, as with GS, previous information on the phenotype-marker linkage or localization of QTLs on the genome and their relative effect on the phenotypic variance becomes unnecessary [212]. The breeding cycle could be significantly shortened, as with GS, marker-based selection is performed on seedlings at a very early stage, eliminating the need for expensive and time-consuming field testing [211]. As for traditional phenotypic selection, to increase GS accuracy, it is advisable to test the population in an environment that is similar to the target environment in which plants are expected to live, in order to avoid genotype x environment interactions [210,213]. GS is currently used routinely for livestock breeding but its application to forest breeding is still very limited. Nonetheless, several works exploring the potential of GS in conifer breeding already exist, mostly focused on traits, such as growth and wood quality. In Pinus taeda, genomic breeding values were estimated on 149 individuals of a cloned progeny from 13 crosses, using 3406 markers simultaneously. The accuracy varied according to the considered trait, ranging between 0.61 and 0.83 for the lignin and cellulose content, while it was lower (0.30 to 0.68) for growth [214]. In Pinus pinaster, 2500 SNPs were used for genotyping 661 individuals from two different generations. The average prediction accuracy using different statistical models and validation scenarios was between 0.4 and 0.5, with small differences according to the considered trait [212]. Other studies evaluated the accuracy of GS in several conifer species, such as Picea mariana [215], Picea abies [216,217] and Picea glauca [218]. Overall, the results obtained with GS in conifers are encouraging, even though significant differences can be found depending on the number of markers [217,219] and types of population used [208,220]. GS studies on DT in trees are still very rare. To our knowledge, the first one was published on a Eucalyptus hybrid population, evaluating 1130 clones with 3303 SNPs [221], but there is no doubt that in the near future research on this area will expand.

## 5. Conclusions and Future Directions

Plant adaptations to DS are likely to become progressively more important in the near future, as most of the recent climate models suggest an increase in the length and intensity of dry periods in many regions [222]. Conifer response to low water availability is a very complex trait, involving a variety of different physiological processes and molecular mechanisms. Nonetheless, in the last decade, substantial progress has been made in understanding drought responses, especially with the availability of new technologies that allow high-throughput phenotyping and genotyping. Many genes and pathways involved in the control of drought responses have been discovered and analyzed in detail, such as those related to ABA, signaling, carbohydrate metabolism and the production of protecting molecules [140]. The availability of a great number of low-cost molecular markers, mostly SNPs, is making it possible to apply new advanced approaches to tree breeding [84], so that techniques, such as GS, could be applied routinely to substantially reduce breeding cycles [211]. However, not all the traits that are thought to be involved in drought resistance have been thoroughly understood, because conifers include a great number of species, both isohydric and anisohydric, so the strategies to adapt and survive in the face of water stress might be different between families, genera and species. Till now, most of the efforts in studying drought responses were performed on species with a high economical value, especially those grown on plantations. This is understandable, but it means that many ecologically important taxa, such as Cupressaceae, should receive more attention, as they could provide valuable information, being remarkably drought-tolerant [46]. Efforts have been made to integrate different approaches to better understand conifer adaptations to water stress [49,66,133,135,145,164]; most of the studies, especially those looking at gene expression, were performed on seedlings or young ramets [142,144,145,150,223]. This, although important to understanding the molecular mechanisms involved in plant responses to drought, does not necessarily reflect the behavior of an adult tree when subjected to the same conditions. It is an example the remarkably different requirements present in several species between seedlings and mature trees in relation to their shade tolerance [224]. Therefore, it would be advisable to also focus on mature plants. Another factor that should be taken into account is the type of DS considered. A single extreme episode of drought is different from a low but constant water availability or multiple stress episodes during the growing season. Plant strategies to cope with different types of stress could be different, so it is important to try to match, as much as possible, the range of conditions that the studied plant may face in the natural environment, as well as in cultivation. Only integrating data from multiple disciplines and considering all the variables that could influence plant drought responses in a natural context, including the type of soil [225,226] and microbiome [227–229], will a more thorough comprehension of this complex trait in conifers be possible.

**Author Contributions:** Conceptualization, P.B. and N.L.P.; methodology, P.B. and N.L.P.; validation, P.B. and N.L.P.; formal analysis, P.B. and N.L.P.; investigation, P.B. and N.L.P.; resources, P.B. and N.L.P.; data curation, P.B. and N.L.P.; writing—original draft preparation, P.B. and N.L.P.; writing—review and editing, P.B. and N.L.P.; visualization, P.B. and N.L.P.; supervision, P.B. and N.L.P.; project administration, P.B. and N.L.P.; funding acquisition, P.B. and N.L.P. All authors have read and agreed to the published version of the manuscript.

**Funding:** This research was funded by FEM projects with the grant numbers P1611006I and P1611034.

**Data Availability Statement:** Not applicable.

**Conflicts of Interest:** The authors declare no conflict of interest.

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
