# Peer review of "Toward the Genetic Improvement of Drought Tolerance in Conifers: An Integrated Approach"

_forests, doi:10.3390/f13122016_

Round 1

Reviewer 1 Report

The paper summarized the different strategies of coniferous forest to deal with drought stress, which has certain reference value for other plants. The overall article is well written and the summary is comprehensive, but it needs to be modified before acceptance:

1 It is suggested that the contents of Section 2 (Drought tolerance) can be integrated into Section 3

2 In the Section 3, the response strategies of coniferous forests to drought are not clearly written. The author should indicate that several strategies are included, and then explain them one by one.

3 It is suggested to draw a model map of the contents of the Section 3 to show the different strategies of conifers to deal with drought.

4 In the Section 4, It is recommended to set subtitles (4.1, 4.2, 4.3) containing three research contents (traits at analytical/molecular, physical/biological and generic level)

5 The author is requested to check the full text and simplify it. There are many repeated expressions in the manuscript

Author Response

REVIEWER 1

Comments and Suggestions for Authors

The paper summarized the different strategies of coniferous forest to deal with drought stress, which has certain reference value for other plants. The overall article is well written and the summary is comprehensive, but it needs to be modified before acceptance:

1.  It is suggested that the contents of Section 2 (Drought tolerance) can be integrated into Section 3

Section 2 was integrated in Section 3.

2.  In the Section 3, the response strategies of coniferous forests to drought are not clearly written. The author should indicate that several strategies are included, and then explain them one by one.

Section 3 was modified and divided in subsections to explain more clearly the different strategies to cope with drought.

3.  It is suggested to draw a model map of the contents of the Section 3 to show the different strategies of conifers to deal with drought.

A schematic representation of the contents of Section 3 was prepared and presented in Fig, 1.

4.  In the Section 4, It is recommended to set subtitles (4.1, 4.2, 4.3) containing three research contents (traits at analytical/molecular, physical/biological and generic level)

Section 4 (Section 3 in the new version) was modified adding subtitles.

5.  The author is requested to check the full text and simplify it. There are many repeated expressions in the manuscript

The text was checked and the main repeated expressions were replaced by acromyms.

Reviewer 2 Report

Climate change is increasing the risk of drought stress in many regions on earth, and I’s important to understand plant response and adaptation mechanisms to drought in order to properly manage tree populations and select those individuals or provenances that show a higher resistance level. In this manuscript, the authors described the main approaches taken in the study of conifer adaptation to low water availability, the advantages and limitations of each, and the main results obtained with each of these approaches in the recent years. The following points need to be considered or improved in this manuscript.

1.    As broad-leaved wood species are important part in global forest ecosystem on earth, and lots of drought stress related studies have been performed, therefor It will be more complete if the authors add a discussion section of the differences between broad-leaved wood species and conifers in response to drought stress.

2.    In Figure1, ‘Next Generation breeding’s integrated processes to improve drought stress in conifers’, More detailed information can be added in image of ‘Next Generation breeding’.

3.    Line 267-269, the genes’ name should be in italic format.

Author Response

REVIEWER 2

Comments and Suggestions for Authors

Climate change is increasing the risk of drought stress in many regions on earth, and I’s important to understand plant response and adaptation mechanisms to drought in order to properly manage tree populations and select those individuals or provenances that show a higher resistance level. In this manuscript, the authors described the main approaches taken in the study of conifer adaptation to low water availability, the advantages and limitations of each, and the main results obtained with each of these approaches in the recent years. The following points need to be considered or improved in this manuscript.

1.  As broad-leaved wood species are important part in global forest ecosystem on earth, and lots of drought stress related studies have been performed, therefor It will be more complete if the authors add a discussion section of the differences between broad-leaved wood species and conifersin response to drought stress.

We agree with the reviewer that broad-leaved trees are an important part of forest ecosystems. Nevertheless, this review is focused on conifers. A comprehensive discussion about the differences between conifers and broad-leaved species would increase the manuscript length considerably and in our opinion it would be out of scope. We have specified this in the text of the manuscript and several references to papers specifically reviewing differences between conifers and broad-leaved plants were added.

2.  In Figure1, ‘Next Generation breeding’s integrated processes to improve drought stress in conifers’,More detailed informationcan be added in image of ‘Next Generation breeding’.

The figure was modified, adding more detailed information.

3.  Line 267-269, the genes’ name should be in italic format.

We could not find the genes’ name indicated by the reviewer. AP2/ERF, bZIP, TCP, WRKY and MYB are not genes’ names but classes of transcription factors.